# Neurons: The Interplay between Cytoskeleton, Ion Channels/Transporters and Mitochondria

**DOI:** 10.3390/cells11162499

**Published:** 2022-08-11

**Authors:** Paola Alberti, Sara Semperboni, Guido Cavaletti, Arianna Scuteri

**Affiliations:** 1Experimental Neurology Unit, School of Medicine and Surgery, University of Milano-Bicocca, 20900 Monza, Italy; 2NeuroMI (Milan Centre for Neuroscience), 20126 Milan, Italy

**Keywords:** cytoskeleton, ion channels, transporters, axonal excitability, biomarkers

## Abstract

Neurons are permanent cells whose key feature is information transmission via chemical and electrical signals. Therefore, a finely tuned homeostasis is necessary to maintain function and preserve neuronal lifelong survival. The cytoskeleton, and in particular microtubules, are far from being inert actors in the maintenance of this complex cellular equilibrium, and they participate in the mobilization of molecular cargos and organelles, thus influencing neuronal migration, neuritis growth and synaptic transmission. Notably, alterations of cytoskeletal dynamics have been linked to alterations of neuronal excitability. In this review, we discuss the characteristics of the neuronal cytoskeleton and provide insights into alterations of this component leading to human diseases, addressing how these might affect excitability/synaptic activity, as well as neuronal functioning. We also provide an overview of the microscopic approaches to visualize and assess the cytoskeleton, with a specific focus on mitochondrial trafficking.

## 1. Introduction

Neurons are perennial cells entitled to signal transmission, and their morphological and functional organization is suited to this task. In fact, neurons have developed a very polarized structure (with different functions for dendrites and axons) and have a key physiological feature, excitability, which allows them to build networks exploited for signal transmission [1]. As perennial cells [2], unable to replicate and stuck in the G0 phase of the cell cycle [3], this polarized structure and the correct neuronal functioning should be tightly controlled and preserved, with multiple checkpoint mechanisms and a relevant energy consumption.

## 2. Neurons and Their Cytoskeleton

Neurons are able to build a very complex and specialized network, where synapses are pivotal for signal transmission, and it is essential that all the cellular components (neurotransmitter vesicles, voltage-gated channels, etc.) produced in the soma can reach dendrites and axons [4]. Since this task is energy-consuming, mitochondria play a crucial role [5]. Moreover, to maintain neuronal functioning each cellular compartment must be easily reached, and the different components of the neuronal cytoskeleton are responsible for this [6,7,8]. Briefly, there are three main components in the cytoskeleton: microtubules, microfilaments, and intermediate filaments [6]. Microtubules are composed by the assembly of dimeric units of α- and β- tubulin that form cylindrical structures of about 25 nm of diameter. Microfilaments are polymers of the monomeric protein actin, with a diameter of about 7 nm, while intermediate filaments differ from one cell to another, and they have a diameter of about 10 nm [7]. These components are responsible for both the structural integrity of the cell and for dynamic events such as axonal transport.

Microtubules and microfilaments are very dynamic structures, able to change their composition and assembly in response to cellular needs, and perturbations in functioning are at the basis of several human diseases, as exemplified in subsequent Section 5.

Both microtubules and microfilaments are polarized polymers with a plus-end and a minus-end, and they are essential for the elongation of dendrites and axons [6,7,8]. Moreover, microtubules are important to determine neurons’ polarity. During the early stages of development, neurons have a rounded shape, which is progressively modified allowing elongation of long axons, and finally the appearance of an arborized dendritic tree [7,8]. Although microtubules support elongation of both axons and dendrites, there is a difference in their organization. In particular, in the axon, the minus-ends of microtubules are always oriented toward the cell body, while the plus-ends toward the periphery. On the contrary, in the dendrites, the microtubule polarity is not so strictly determined [6,7,8]. The different microtubule organization contributes to determining the fate of the neurite (being a dendrite versus an axon) [9,10]. Actin microfilaments concur with this process too [6,7,8,11]. The microfilaments network is a very dynamic structure composed by actin polymers, able to interact with different associated actin binding proteins. They enable the formation of two different structures: actin-rings and actin patches [8]. Ring-like structures are responsible for the mechanical support and for the distribution of important membrane proteins, such as voltage-gated channels [12] and ionic pumps, in order to create an electrochemical gradient between the two sides of the plasmalemma. On the contrary, actin patches seem to be important in anchoring vesicles and organelles [13].

Besides neuronal shape and polarity, microtubules are also in charge of transporting molecules and organelles from the cell body to the periphery (anterograde transport), and vice versa (retrograde transport) [14,15,16]. Anterograde transport depends on kinesins, motor proteins that move toward the plus-end of microtubules and it is crucial to the distribution of proteins customized at the level of the soma (such as neurotransmitters and other synaptic components) towards the periphery; it can be distinguished into fast and slow types [14]. Dynein is the motor protein responsible for retrograde transport, which is exclusively a fast transport which allows the neuron body to receive stimuli coming from its terminal endings [14]. The retrograde transport is also involved in neuronal survival, allowing a feedback signal to the neuron body via neurotrophic factors [14,15]. Table 1 gives an overview of the cytoskeleton’s role in protein trafficking.

The cytoskeleton is essential for the proper functioning of mitochondria [4,5], that are crucial elements for neurons, which are high energy-demanding cells. Mitochondria are pivotal for cell bioenergetics, and in fully differentiated neurons they are entitled to efficiently produce ATP via OXPHOS, thanks to ETP and ATP synthase [22]. However, mitochondria are also involved in other fundamental neuronal processes, such as calcium signaling, trafficking and apoptosis, exploiting their communication with endoplasmic reticulum and lysosomes [23]. To fulfil their role, mitochondria are entitled to a lively and dynamic trafficking, which makes their interaction with the cytoskeleton fundamental. In particular, axons and the presynaptic area have relevant energetic demands and require a balanced regulation of calcium ions, provided by mitochondrial trafficking and functioning [24]. To allow mitochondria to move back and forward, or even stop in a precise area [5,25], the role of microtubule motors is essential, and they are scaffolded on mitochondria via the interaction between Miro proteins and the motor-biding proteins TRAK and metaxins (MTX) [24,26]. Miro proteins are calcium-sensitive GTPase which are located in the outer membrane of mitochondria and are in charge of both long-distance mitochondrial transportation and mitochondrial anchoring in sites where high local energy levels are required [27] TRAK1 and TRAK2 are dynein-activating adaptors [26,28,29] which are also able to bind kinesin [26,30] and it was shown in vitro that they are able to activate kinesin and dynein motility [26,29,31,32]. MTX are proteins located in the outer mitochondrial membrane that were first described as being involved in protein importing in mitochondria [33,34]; however, it was later demonstrated that they are also a core component of mitochondrial adaptor complexes [35]. In line with this observation, TRAK1/2 or MTX1/2 knockdown in human neurons alters both anterograde and retrograde mitochondrial motility [35]. If transportation is important, the same is also true for mitochondrial docking, particularly in the presynaptic sites [5,25]. Local anchoring is obtained via associations with the microtubule or actin cytoskeleton [36,37]. At the presynaptic level, syntaphilin can stop mitochondrial transportation, acting as a microtubule-binding protein, either associating with mitochondrial outer membrane and/or sterically inhibiting kinesin interaction with microtubules [36,37]. Moreover, local energy demands can promote mitochondrial halting: if there is a local increase in glucose concentrations, TRAK 1/2 O-GlcNAlation takes place resulting in TRAK1 interaction with the acting-binding protein FHL2 [38]. Another possible mechanism is related to the effects of calcium on Miro; calcium binding reduces transport, even if the exact sequence of actions that lead to this effect is still a matter of debate [39]. These strategies to mobilize/dock mitochondria are not a specific feature of mitochondria, but it is part of the general mechanism neurons undertake to mobilize vesicles [40] and organelles; similar mechanisms were in fact described for other subcellular components as observed in old studies on the giant squid axon [41,42,43].

## 3. Cytoskeleton, Neuronal Excitability and Ion Channels

As anticipated, one of the key neuronal features is the ability to transmit a signal. Briefly, a stimulus, able to evoke an action potential (AP) determines a membrane voltage gradient able to propagate through the axon thanks to stereotyped phenomena occurring with a perfectly timed and ordered activation of voltage-gated ion-channels (i.e., sodium and potassium ones), at the cost of energy granted by the transmembrane ATPase Na+/K+ pump [44]. The membrane gradient potential made up by the Na+/K+ pump is the essential precondition for signal transmission, together with the constant availability of ATP molecules, as an ATPase pump. Since neurons cannot stock glucose (differently, for instance, from muscle cells which can store it as glycogen), they are strongly dependent on the blood flow to obtain ATP. Moreover, to maximize the ATP production from glucose, neurons must exploit oxidative phosphorylation, meaning that they also need both O_2_, again through the blood flow, and a high number of mitochondria to achieve an adequate level of energy production [44]. Therefore, for signal transmission, the aforementioned role of the cytoskeleton in supervising mitochondrial functioning and axonal transport is crucial. However, the cytoskeleton plays an even more dynamic role in neuronal transmission given some notable interactions with ion channels and receptors involved in AP generation/propagation and synaptic transmission. In particular, actin is a pivotal element not only for structural stability but also for the regulation of ion channels, transporters, and receptors modulating their expression on cellular surface and/or intracellular trafficking [45]. Notably, the relationship works also on the reverse: several ion channels have been demonstrated to influence cortical actin networks (e.g., voltage-gated potassium channels) [45]. An overview of the basic element and mechanisms related to neuronal excitability is presented here, laying the ground for the exploration in future research related to human diseases, as better exemplified in Section 5. Ion channel trafficking and clustering is crucial in the generation of AP, in particular at the axonal initial segment (i.e., a 10–60 um unmyelinated located at the proximal axon/soma interface) and at the node of Ranvier, as well as the synaptic membrane. In fact, the axon hillock is not involved in AP initiation and, therefore, the spatial localization at the level of the axonal initial segment and at the node, guaranteed by the mechanical and transport functions of the cytoskeleton network, plays a pivotal role in the generation of the AP. Ankyrins and actin-spectrin cytoskeleton are in charge of this task, with a contribution from the adaptor proteins (e.g., ankyrin G, syntrophin) and adhesion molecules [46,47]. As sodium and potassium voltage-operated ion channels (Na-VOC and K-VOC, respectively) are pivotal in AP generation [48], a more detailed insights on their relationship with the cytoskeleton is necessary. Na-VOC channel clustering, in sites of AP initiation, is established by specific membrane cytoskeleton actin-spectrin modules [49,50,51]; the former is linked to an adaptor, the ankyrin-G, that connects the actin-spectrin modules to Na-VOC channels, neurofascin 186 (a L1 family cell adhesion molecule), and K-VOC subunits, KCNQ2 and KCNQ3 [52,53,54]. The presence of KCNQ2 and KCNQ3, guaranteed by the cytoskeleton, is fundamental to modulate excitability at the axonal initial segment and at the node of Ranvier [55,56,57]. The relevance of Na-VOC and K-VOC localization and, therefore, their correct functioning in these sites, is supported by the observation that mutations leading to altered Na-VOC and K-VOC channels at axonal initial segment disrupt neuronal excitability; a fair example of this dysfunction is given by recurrent epileptic seizures in case their anatomical configuration is altered [58,59]. Data emerged from in vitro and in vivo studies elucidated the molecular mechanisms by which ankyrin-G clusters mammalian K-VOC and Na-VOC, the latter being characterized by cytoplasmic anchor motifs which are crucial for their localization at the axonal initial segment [53,60,61]. Ankyrin-G is characterized by a membrane interaction domain (24 solenoidal ankyrin repeats are described) and it was suggested via mutagenesis experiments that ankyrin-G repeats 13–15 are responsible for the interaction with the Na-VOC anchor [60,61,62]. A similar interaction was suggested for K-VOC3 anchor sequences [50,54,58]. Most notably, in the peripheral nervous system, clustering at nodal level has been strongly linked to gliomedin secreted by Schwann cells, and the axonal cell adhesion molecule neurofascin-186 [63,64]. Only after this initial event, cytoskeletal and scaffolding proteins ankyrin, as well as βIV spectrin, are recruited to nodes to allow channels binding necessary for AP [60]. The axonal initial segment molecular composition is fairly similar to the node and these similarities suggested that their formation might be the same [65], with the difference that in this case the extracellular matrix is thought to contribute, thanks to a specialized brevican-containing matrix. Alternatively, ankyrin-G was once again suggested as crucial [53,60,61], and its role is further supported by experimental evidence since ankyrin-G lacking mice showed a deranged axonal initial segment ion channel clustering in Purkinje cells [53,66]. Nevertheless, other components seem to be implicated since βIV spectrin loss was also linked to altered axonal initial segment clustering [67].

Apart from Na-VOC and K-VOC, other channels/transporters have a lively relationship with the cytoskeleton. N-methyl-D-aspartate receptors’ (NMDARs) interaction with actin is pivotal in synaptic plasticity, which is a pre-requisite, for example, for learning and memory [68]. Another example is given by voltage-gated calcium channels: actin cytoskeleton participates in the crosstalk of L-type of these channels and mitochondria, obtaining a regulation of noradrenergic activity in the locus coeruleus, a fundamental relationship for integrative brain functions [69]. Transient receptor potential (TRP) cation channels, which are involved in Ca2+ signaling, also have a peculiar relationship with the cytoskeleton which might act as a modulator of TRP, but are modulated by TRP via a downstream effect of TRP function [70]. Therefore, the cytoskeleton, and actin in particular, may represent a underrecognized mechanism of modulation and activation of voltage-gated and non-voltage-gated ion channels in neurons [45].

## 4. Microscopic Approaches to Visualize and Assess Cytoskeleton Function

Th cytoskeleton can be assessed microscopically with many different approaches developed over the years. An overview is given here to present the possible techniques that could be relevant when designing a research plan aiming at ascertaining the role of a specific cytoskeleton component and/or cytoskeleton functioning. Light microscopy was the key technique in earlier studies of cytoskeleton components. The first studies date back to the 17th century and they generically defined cytoskeleton proteins as a network of neurofibrils [71]. However, it was not until the 1940s, when immunolabeling with fluorescent antibodies for light microscopy was introduced [72], that more refined observations were possible. These were first used to study neuronal cultures and only later specifically to characterize neuronal cytoskeleton. Antibodies against tubulin and actin (or fluorescent phalloidin) were introduced to label microtubules and actin filaments in the processes of neuroblastoma [73]. To highlight the relevance of this advancement, it should be noted that the restriction of the microtubule-associated protein MAP2 to the somatodendritic compartment was first demonstrated via tubulin immunolabeling [74]. However, it was only at the end of the 1990s that fixation and staining protocols had reached the final refinement setting and fluorescent phalloidin, anti-tubulin, and MAP2 antibodies were used as standard labelling methods [75].

However, light microscopy did not allow an in-depth analysis of the cytoskeleton due to the intrinsic physical limitation related to the diffraction limit (200 nm). A fundamental turning point in the description and comprehension of cytoskeleton structures was represented by the availability of transmission electron microscopy (TEM) [76]. Since it uses electrons, rather than light, TEM can overcome the diffraction limit, thus allowing the actual description of neurofilaments and microtubules. TEM also permitted the description of actin filaments and the axon initial sequence [77]. Unfortunately, even if TEM has contributed, and still contributes, to the investigation of the neuronal cytoskeleton, it has some well-known limitations. These are primarily represented by the technical challenges posed by the fixation process, which can damage these subcellular components [78,79,80]. Furthermore, proteins associated with the cytoskeleton are not that easily detectable using TEM. Over the years, technical advancements had allowed the amelioration of morphological studies in this field; in particular, cryogenic electron microscopy represented a relevant improvement in preserving samples, since through a rapid freezing process the natural state of the molecules is better preserved [81,82]. Moreover, immune-electron microscopy allowed better characterization of single components of the cytoskeleton, even if the specimen preparation requires a careful and well-conducted procedure.

Subsequently, the development of fluorescent microscopy, and in particular of confocal microscopy, increased our ability to resolve the cytoskeleton, since cytoskeleton-binding fluorescent antibodies allow the examination of the cytoskeleton structure, motility, organelles organization and trafficking along the microtubules. However, this technique also has some pitfalls that should be acknowledged when selecting a specific experimental design on the basis of the scientific hypothesis to be tested. When fixation is performed, potential artifacts should also be weighed in this case. Moreover, the cytoskeleton is very packed, making difficult to discriminate between single structure, also considering, again, the diffraction limit of this kind of microscopy.

Since the neuronal cytoskeleton is a dynamic, and not a static complex, an intriguing advancement in its study was represented by live imaging. Fluorescein phalloidin, a high-affinity F-actin probe conjugated to the green fluorescent dye, is the gold standard for the evaluation of actin filaments in live cells [83], while, for the evaluation of microtubule dynamics, the end-binding protein (EB1 and EB3) involved in the growth of microtubules’ plus-end, tagged with a green fluorescent protein [84], can be used. Even in these cases, however, there are difficulties to overcome. The probe could be toxic, or it could be bundling, somehow altering the polymerization/depolymerization dynamics of the various structures of the cytoskeleton or modifying the binding site of proteins associated with the cytoskeleton [85,86,87,88].

Over the last decade, many techniques at the forefront super-resolution allowed us to further deepen our understanding of the cytoskeleton and the dynamics between its components and the associated proteins. All the super-resolution techniques share the capacity to break the light diffraction barrier, reaching, in some cases, resolution limits down to a few tens of nanometers [89]. They can generally be divided into two major groups: (i) patterned light illumination techniques with stimulated emission depletion (STED) and structured illumination microscopy (SIM), and (ii) localization-based techniques with stochastic optical reconstruction microscopy (STORM) and photoactivation localization microscopy (PALM). Each of these techniques has allowed the characterization of important aspects of cytoskeletal organization: SIM has been used to detect dynamic changes in cytoskeleton architecture) [90], while STORM, using a phalloidin probe, permits the reconsideration of actin organization in axons, where a ring-like structure wraps them every 180–190 nm, but not in the dendrites [12]. The same organization can be detected with STED in living neurons [91]. STORM was also used to underline the dynamics of neurofilaments in the axons, showing that neurofilaments, even if they have been considered stable structures, also undergo continuous turnover based on actin-dependent retrograde flow and anterograde and retrograde microtubule-dependent transports [92,93]. With PALM technique, it has been possible to visualize microtubule organization and to ascertain their role in organelle movement in living cells [89]. These techniques, together with the development of specific (and also transgenic) probes for the cytoskeleton, will allow us to investigate the cytoskeleton with unprecedented spatial and temporal resolution, thus pushing the limit forward to the nanoscale. Table 2 summarizes the main super-resolution microscopy techniques characteristics, with their advantages and disadvantages.

Among the super resolution techniques, STED might greatly benefit from the use of transgenic approaches. STED nanoscopy enabled a super resolution fluorescence microscopy in living brain slices [98,99] and in the brains of anesthetized live mice [100,101,102,103], using the overexpression of fluorescent proteins to mark the actin network. However, this procedure increases the risk of tissue photodamage. To overcome this possible bias, transgenic animals can be used, as demonstrated by Masch et al. [104], and the molecular specificity of a genetically encoded self-labeling enzyme tag can be exploited after fusion with the protein of interest, combining it with the superior photophysical performance of organic fluorophores.

Concerning the cytoskeleton function, there is increasing interest regarding its involvement in organelle transport, in particular mitochondria. Light microscopy with the use of fluorescent probes can evidence how mitochondria move along axons [105,106,107]. The fluorescent probes are inserted in the cells under investigation by transfection or MitoTracker dye, an easy and ready-to-use probe that accumulates in mitochondria through membrane potential. When analyzing mitochondrial movement along the axon, the percentage of stationary and in-motion mitochondria must be taken into account, since typically 60% are stationary [108,109]. In-motion mitochondria are nearly equally divided between those that move anterogradely or retrogradely. However, in the visualization of mitochondrial trafficking, a high-throughput, dynamic and long-term analysis of the cellular events can be achieved via holotomographic microscopy, which enables non-invasive visualization and quanti-fication of living cells without compromising cell integrity. Holotomographic microscopy is a novel approach that combines holography and tomography technologies, enabling the study of cells and thin tissue slices via three-dimensional images [110]. These images are obtained by exploiting the specific refraction index (RI) of each structure/cell, an intrinsic optical parameter that is pivotal for this technique. The holographic technique exploits two major light parameters (i.e., amplitude and phase) and captures light information scattered from each sample at once; therefore, an optical phase delay takes place when light passes through the object under study, which is highly related to the specimen thickness and to the distribution of its RI values. The holography technique measures this phase delay, giving an indirect clue as to the object’s inner composition [111]. The next step is the optical diffraction tomography that reconstructs the 3D distribution of RI values; this returns the structural and chemical characterization of the specimen. As a consequence, a quantitative and label-free analysis of volume and subcellular components on live unstained cells and tissue slices is possible [112,113], and dry mass, morphology, and dynamics of the cellular membrane, as well as protein content, can be evaluated too [114]. Holotomographic microscopy can be particularly suitable for mitochondrial trafficking studies, allowing a label-free alternative, ensuring the capture of the behavior of mitochondria at high frequencies and for unlimited periods of time. Of course, it is not a direct visualization of the cytoskeleton, but it is a simple and indirect measure of its functioning via one of its essential tasks, i.e., mitochondrial motility. On the other hand, the absence of specific probes could make it harder to clearly identify subcellular structures (unless they are easily recognizable, such as lipid droplets). In Figure 1, we present selected time frames of Appendix A that we obtained as representative images of a physiological mitochondrial motility study obtained with this approach. Mitochondria (shown in bright white and indicated by white arrows in the image) were recognized via their RI and followed (e.g., tracked) over time and space in our sample.

## 5. Examples of Human Pathology Related to Cytoskeleton Dysfunction

Given the high relevance of the cytoskeleton for neuronal physiology and survival, it is far from surprising that its perturbations can cause human diseases. Here, we will present some significant examples highlighting the role of cytoskeleton alterations in epilepsy, cognitive dysfunction and intellectual disability, and peripheral neuropathies (PNs).

### 5.1. Epilepsy

Epilepsy is the first condition that can be easily associated with a disruption of one of the fundamental properties of neurons, i.e., excitability and the fine tuning that should be maintained over it. Epilepsy is defined as a propensity to develop seizures, which are abnormal, hypersynchronous discharges of cortical neurons [115]. Worldwide, it is estimated that 50 million people are affected by this condition, affecting both children and adults [116]. Ion channels are certainly the simplest targets to think of as the *primum movens* of epilepsy, given their role in AP generation, and, in fact, many antiepileptic drugs (AEDs) are ion channel modulators [117,118]. However, the neuronal cytoskeleton should also be considered, and microtubules are key players in the bioelectric properties of neurons since they can resemble wires, able to transmit and amplify electric signals via the flow of ions [119], and, therefore, they might be a novel druggable target and a specific molecular component to be investigated in pathogenetic studies. Epilepsy can arise as a consequence of altered neuronal development (for example related to microtubules’ dysfunction) or altered axonal transport, and cytoskeleton modulation might be, in the next years, a possible strategy to modulate altered neuronal excitability.

Despite the fact that the effects of microtubule dynamics on neural excitability and synaptic transmission have been suggested in the past [120,121], only more recently have microtubule dynamics been more closely linked to the generation of chronic epilepsy [122]. Altered neuronal excitability can be due to microtubule impairment during the development of synapses, in which finely regulated mechanisms such as phosphorylation, acetylation, tyrosination, and glutathionylation are necessary [123]. When alterations take place in this process, the result is a perturbed neuronal network, leading to cognitive deficits and neurodevelopmental disorders such as microencephaly, lissencephaly, polymicrogyria, and infantile epileptic syndromes [123,124]. Therefore, microtubule functioning is a possible pathway to be investigated in models of hereditary epilepsy. On a similar ground, another condition can also be enlisted, even if not strictly for its direct link between hyperexcitability and the cytoskeleton: the juvenile myoclonic epilepsy. This disorder is related to mutations of the EFHC1 protein, which is crucial in cell division, accumulating at the centrosome during interphase. A colocalization of EFHC1 with alpha-tubulin at the mitotic spindle occurs, suggesting a possible interaction with microtubules during brain development leading to neurological disturbances typical of this syndrome. Moreover, in neurons, EFHC1 is localized at soma and dendrites and its overexpression in mouse hippocampal primary culture neurons induces apoptosis [125]; this pro-apoptotic effects of EFHC1 is linked to an enhanced Ca2+ influx through R-type voltage dependent Ca2+ channel (Cav2.3).

Considering axonal transport, there is a specific family of proteins that can predispose to epilepsy: kinesins. They undergo modifications during neurodevelopment leading to hyperexcitability in the case of loss of neuronal kinesin heavy chain KIF5A. Xia et al. described effects on excitability in Cre-recombinase transgene mice with direct inactivation of KIF5A in neurons postnatally (KIF5A mutants die immediately after birth): 75% of mutant mice exhibited seizures and died at 3 weeks of age [126]. Nakajima et al. were able to link KIF5A alterations to seizures onset, in fact, due to impaired GABA receptor-mediated synaptic transmission, causing loss of the inhibitory regulation KIF5A has on neural transmission [127].

Notably, post-natal changes in the cytoskeleton might also play a pivotal role in epilepsy. As an example, brain-derived neurotrophic factor (BDNF, a neurotrophin able to actively modulated the mature nervous system [128]) alterations were related to seizure onset due to possible effects on microtubules. BDNF is suggested to be released into mature synapses during theta activity with local effects on actin cytoskeleton, similar to those that take place during neuronal development [129]. However, the exact role of BDNF in epilepsy is not fully understood. There is conflicting evidence on the role of BDNF, since it was described as enhancing neuronal survival and growth factor in epileptic rats [130], but it has also been implicated in the pathogenesis of hippocampal hyperexcitability and epilepsy when over-expressed [121].

The crucial role of microtubules in epilepsy is further evidenced in experimental studies showing the positive effect of microtubule-modulating agents on seizures. Xu and collaborators [122] studied microtubule dynamic in 30 human epileptic tissues and in two different chronic epilepsy rat models. In human specimens Tyr-tubulin presented an elevated expression in the neocortex and microtubule dynamics was dysregulated. The same alterations were found in the rat model where it was also demonstrated that the administration of a microtubule modulating agent (noscapine, which reduces the dynamics of microtubules by increasing the stopping time of microtubules [131]) attenuated the progression of chronic epilepsy and, by contrast, the microtubule-depolymerizing agent colchicine aggravated the progression of chronic epilepsy. Wu and colleagues [132], in a rat model of recurrent epilepsy, observed both α- and β-tubulin downregulation, as well as hippocampal neuron loss, in animals with recurrent seizures, thus reinforcing the role of microtubule system in epilepsy. The presented examples, therefore, support, once again, the idea that for a proper neuronal functioning (e.g., excitability) physiological microtubules are a crucial and not optional.

### 5.2. Intellectutal Disability (ID)

Another nosographic entity where cytoskeleton alterations may play a crucial and still underexplored role is ID, which is defined as a pathological condition characterized by limited intellectual functioning and adaptive behaviors, affecting 1–3% of the world’s population, lacking of an efficacious pharmacological treatment [133]. Further investigations on the role of the cytoskeleton in this regard may shed light not only on the mechanisms leading to this condition, but also define potential precision medicine strategies (e.g., targeted therapies). Similar to epilepsy, a crucial step in ID is neurodevelopment, and the cytoskeleton (particularly the actin component) can, of course, be one of the key players. Cytoskeleton dynamics, whose regulation depends on Rho GTPases transduction, has been linked to ID. In particular, cytoskeleton alterations during neuronal development leading to altered neuronal migration, neuritogenesis, and synaptic plasticity were demonstrated in Down syndrome (DS), Rett syndrome, Fragile X syndrome (FXS), and autism spectrum disorder (ASD). In DS patients, the regulators of actin filament polymerization and branching, CDC42 and WASP, have been implicated as relevant, conditioning alterations in protein–protein interaction, leading to ID [134]. Actin polymerization was reduced, with the possible result of altered dendritic spine morphology and density [135]. In line with this observation, the triplication of the DYRK1A gene encoding proline-directed serine/threonine kinase, located in the critical region of Down syndrome (DS), causes cytoskeleton alteration (primarily involving actin) which was linked to ID [136]. Similarly, actin regulation was demonstrated as being altered in FXS, with an impairment in cytoskeleton constitution [137]. In Rett syndrome, transcriptomic analysis was carried out on preclinical models evidencing dysregulation in cytoskeleton dynamics, actin polymerization, and focal adhesion [138]. Finally, in ASD, a dysfunction of Rho GTPases signaling was described as relevant in its pathogenesis, and nearly 20 genes encoding Rho GTPases regulators and effectors have been reported as ASD risk factors [139]. On a similar ground, actin regulation in stem cells from ASD patients demonstrated altered dynamics of filament reconstruction upon activation of the Rho GTPases RAC1, CDC42, or RHOA, resulting in shorter and less arborized neurites [140]. Therefore, further studies aiming at exploring actin/cytoskeleton modulation in ID field might be suggested.

### 5.3. Peripheral Neuropathies (PN)

PN is a large category under which different diseases affecting the peripheral nervous system are grouped together, encompassing both hereditary and acquired conditions. On the basis of the affected components, patients can develop dysfunction related to sensory (i.e., sensory perception alterations and/or neuropathic pain), motor (i.e., weakness and wasting) and/or autonomic dysfunctions. In hereditary PNs, alterations in cytoskeletal components are a relevant occurrence, related to malfunctioning of cytoskeletal proteins themselves. Charcot–Marie–Tooth (CMT) disease is the most common inherited PN. Genetic and phenotypical heterogeneity is high, but CMT can be mainly divided into more common sub-forms based on the clinical and instrumental findings: demyelinating type (CMT1), axonal type (CMT2), and intermediate types (CMTI) [141]. We list here some examples of CMT in which the cytoskeleton is involved in disease onset. Tubulinopathies (i.e., mutations in beta-tubulin) are related to mutations of TUBB3 genes, leading to a reduction in kinesin localization to microtubule plus-ends, altering axonal transport, and microtubule overstabilization and increased depolymerization [142]. The formin family can also be enlisted (they are proteins involved in direct the elongation of unbranched actin filaments) [143]: when INF2 gene is mutated, a dominant-CMTI type E-form can arise [144]. Mutations in the NEFL gene can lead to several different axonal CMT subtypes (CMT2). Both dominant conditions causing aggregation and recessive mutations resulting in loss-of-function were described [145,146,147], and the affected domains can vary resulting in different pathological effects. Small heat shock proteins (sHSP), HSPB1 in particular, are associated with cytoskeletal abnormalities related to neuromuscular diseases [148,149]. Dominant mutations in HSPB1 are associated with CMT type 2F and distal hereditary motor neuropathy (dHMN) [150]; the S135F mutation leading to a higher affinity of HSPB1 for tubulin overstabilizing microtubules determines neuropathies affecting axonal transport [150].

Another relevant nosographic entity linking PNs and cytoskeleton alteration is related to iatrogenic neuropathies due to chemotherapy agents (the so-called chemotherapy-induced peripheral neurotoxicity, CIPN), that once again shows the pivotal role of cytoskeleton functions in neuronal survival. Among the different chemotherapy drugs, there are some classes that primarily target the cytoskeleton: vinca alkaloids, taxanes, epothilones and proteasome inhibitors. Although CIPN mechanisms are still not completely understood [151] for the aforementioned drug classes, cytoskeleton disturbances have been suggested to be pivotal. Proteasome inhibitors (most notably bortezomib) lead to increased tubulin polymerization resulting in microtubule stabilization that impairs neuronal functions [152]. Vinca alkaloids were shown to be associated with dysfunctional axonal transport, presumably because they prevent microtubule polymerization [153]. Instead, taxanes and epothilones can impair the neuronal metabolism acting with an opposite mechanism, i.e., the hyperstabilization of microtubules [154,155].

## 6. Cytoskeleton Components as a Possible Biomarker for Human Pathology? The Virtuous Example of Neurofilament Light Chain

Cytoskeleton components can also be relevant in biomarker discovery, since, as neuronal disturbances are present, it is likely that key structural elements of the cytoskeleton are released by injured neurons and undergo detectable changes in their biological fluid levels. Among cytoskeleton components, neurofilaments (Nf) have raised interest in the last few years. Nf are part of the class IV of the intermediate filaments, and comprehends Nf light chain (NfL), Nf middle chain (NfM), Nf heavy chain (NfH) [156,157]. Nf are sparse and tortuous in the soma and in the dendrites, whereas they are numerous, mainly straight, long and unbranched in axons [158]. Notably, Nf are even more abundant in large myelinated axons and are highly organized [159]. Given their spatial distribution in neurons, Nf can be released from damaged or diseased axons in significant amounts into blood, and cerebrospinal fluid or serum could be therefore proposed as potential biomarkers of axonal injury, axonal loss, and neuronal death. Among the different types of Nf, NfL seems to be the most promising as potential biomarkers, since they are released by damaged central and peripheral nervous system axons in interstitial fluids [160] and the development of the ultrasensitive Simoa technique allows us to easily measure their levels not only in the cerebrospinal fluid, but also in more convenient serum/plasma samples [161,162]. NfL were already tested in various neurological disturbances affecting both central and peripheral nervous systems [160,161,162,163,164,165,166,167,168,169], even if they have not yet entered in routine clinical practice. As a significant example of the virtuous exploitation of NfL testing that might reach a clinical use, we will rely on a specific disease, i.e., CIPN. To verify that a sound biological rationale was present, NfL levels were initially tested in animal models with promising results. First, NfL were tested in a rodent CIPN model of vincristine-induced neuropathy: a steady, and significant increase during the course of chemotherapy administration (1 month) was demonstrated, leading to a final four-fold increase compared to control animals. This remarkable increase in NfL levels corresponded to a severe axonopathy in peripheral nerves as well as to a loss of intraepidermal nerve fibers, matched by behavioral, neurophysiological alterations [164]. Subsequently, to further explore NfL use in detecting axonal damage, the same group compared serum NfL levels in animals chronically treated with cisplatin and paclitaxel; the former drug primarily targeting the dorsal root ganglia neurons with axonal damage as a secondary event, whereas the latter induces primary axonopathy [154,170]. Serum NfL measurements correlated with the different severity of axonal damage in the two selected models, since both drugs induced serum NfL increase, but higher levels were observed in the paclitaxel-treated group; moreover, earlier alterations were seen in this group, with the magnitude of the NfL increase matching the different severity of morphological and functional alterations. On the basis of these results, NfL were then tentatively tested in a clinical setting and very recently some groups suggested their use at the bedside. Karteri et al. [171] prospectively followed up 59 patients treated with paclitaxel measuring serum NfL before starting chemotherapy, after 2 and 3 weekly courses, and at the end of chemotherapy (12 weekly cycles in total), a significant longitudinal increase was observed and NfL levels were significantly higher in patients affected by more severe toxicity. Huehnchen et al. [172] monitored NfL levels in a cohort of 31 patients treated with paclitaxel and compared NfL levels with healthy control (n = 6) and breast cancer patients not treated with chemotherapy (*n* = 25); they observed a specificity and sensibility higher than 80%, giving promising results even if the low number of subjects included in the analysis make a validation necessary in larger cohorts. In any case, this course of action, from bench to bedside with a strong sound methodological approach, is the one to be followed to evaluate and validate other biomarkers that rely on the fact that many human diseases are due to alterations related to the cytoskeleton.

## 7. Concluding Remarks

The lively role of the cytoskeleton in neurons makes its components crucial both in physiological and in pathological settings. Therefore, they can be both a matter of investigation for studies aiming at dissecting diseases mechanisms, but also for the identification of reliable, clinically-relevant biomarkers for human pathologies, as well as the identification of new, potentially druggable targets.

## Figures and Tables

**Figure 1 cells-11-02499-f001:**
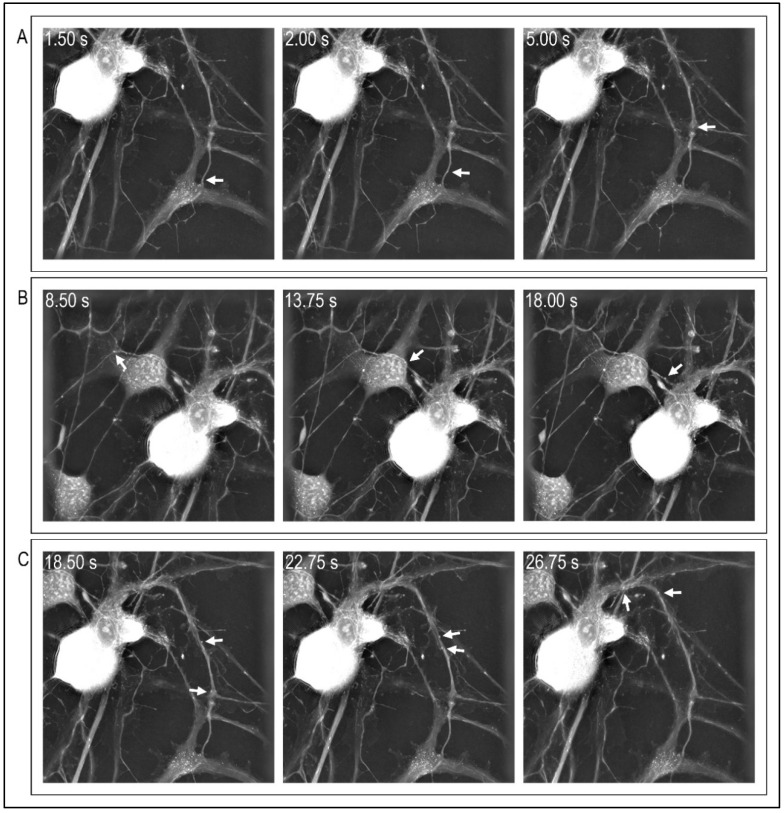
Mitochondrial trafficking with holotomographic microscopy. Each panel (panel (**A**–**C**)) enables us to follow different mitochondria at different time frames (time frame is stated in the upper left corner of each image). White arrows point out single mitochondria in each frame (images obtained with Nanolive holotomographic microscope).

**Table 1 cells-11-02499-t001:** The role of the cytoskeleton in protein trafficking and anchoring.

Class of Transport Protein	Transport	Substrates
**Myosin**	They move specifically along actin filaments. They are involved in contractile forces and short-range transport [17].	Myosin Va is involved in localization of proteins to the somatodendritic compartment and in regulating transport of mRNAs, dense core vesicles, mitochondria and neurofilaments. It is involved in the localization of axonal proteins and modulates mitochondrial movements. More study is needed to fully elucidate these roles and determine their functional significance [18].
**Dyneins**	They move along the microtubule cytoskeleton to facilitate long-range transport [17]. Retrograde transport [19].	It requires the dynein activator, dynactin, that binds directly to dynein and also binds directly to microtubules via Cytoskeletal Associated Protein-Glycine-rich domain [20,21].
**Kinesin**	They move along the microtubule cytoskeleton to facilitate long-range transport [17]. Anterograde transport [19].	Kinesin-1 family drive the transport of a wide range of cargos, including vesicles, organelles, proteins, and RNA particles. Kinesin-2 family drive fodrin-positive plasma membrane precursors, N-cadherin and β-catenin, choline acetyltransferase, and Rab7-positive late endosome-lysosome compartments. Kinesin-3 family drive the motility of synaptic vesicle precursors and dense core vesicles [20,21].

**Table 2 cells-11-02499-t002:** Super Resolution techniques overview.

Super Resolution Techniques	Resolution	Further Insights into	Advantages	Disadvantages	References
**SIM ^1^**	100–250 nm	Dynamic changes in cytoskeleton architecture.	Conventional fluorophores Multiple fluorophores	Reduced resolution. Longer processing time.	[90,94,95]
**STED ^2^**	50–80 nm	Cell matrix interactions. Periodic ring structure of actin in neurons. Actin dynamics in dendritic spine in living neurons. Intermediate filaments network. Microtubule in primary cilia.	Generic dye Speed of acquisition No need for complex analysis	Photobleaching. High-energy laser, often not compatible with living cells.	[90,94,96]
**STORM ^3^**	20–50 nm	Actin organization in cytoskeleton. Cell junctions (integrin keratin, plectin). Microtubule organization.	Very high resolutionSingle fluorophore, specific	Low temporal resolution. Low acquisition speed. Phototoxicity. Complex post-acquisition. Need of special fluorophores.	[90,94,96,97]
**PALM ^4^**	20–50 nm	Mitochondrial proteins. Microtubule organization and role in organelle movement in living cells. Interaction with kinesin motor proteins in neurons.	Very high resolutionSingle fluorophore, specific	Low temporal resolution. Low acquisition speed. Phototoxicity. Complex post-acquisition. Need of special fluorophores.	[90,94,97]

^1^ Structured Illumination Microscopy. ^2^ Stimulated Emission Depletion. ^3^ Stochastic Optical Reconstruction Microscopy. ^4^ Photoactivation Localization Microscopy.

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
