# Peer review of "Neurons: The Interplay between Cytoskeleton, Ion Channels/Transporters and Mitochondria"

_cells, 2022, doi:10.3390/cells11162499_

Round 1
Reviewer 1 Report
The manuscript from Alberti et al it is a well written and interesting review focused on the important role of cytoskeleton in neuronal function.
I do not have any major concern, however I would like to suggest to describe in more details the technique the author mentioned in their work and used to produce the video and images shown in Figure 1.
For example, what distinguishes this technique, olotomography, from others? what is shown in figure 1 exactly? how are mitochondria imaged to produce the white signal? A brief description of the method would be useful for readers that are not familiar with the technique.
Author Response
We are most thankful for this comment.
We updated the 4th section of the review giving a better overview (line 288-316).
"This is a unique technique that enables non-invasive visualization and quantification of living cells without compromising cell integrity. Holotomographic microscopy is a novel approach that combines holography and tomography technologies, enabling the study of cells and thin tissue slices via three-dimensional images[106] [107]; these images are obtained exploiting the refraction index (RI), an intrinsic optical parameter that is pivotal for this technique. The holographic technique exploits two major light parameters (i.e., amplitude and phase) and captures light information scattered from each sample at once; therefore, an optical phase delay takes place when light pass through the object under study which is highly related to the specimen thickness and to the distribution of its RI values. The holography technique measures this phase delay, giving an indirect clue of the object inner composition[108]. The next step is the optical diffraction tomography that reconstructs the 3D distribution of RI values; this gives back the structural and chemical characterization of the specimen. As a consequence, a quantitative and label-free analysis of volume and subcellular components on live unstained cells and tissue slices is possible [109,110]. Notably, these parameters are characterised: dry mass, morphology, and dynamics of the cellular membrane; last, but not least, protein content could also be evaluated too[111]. Holotomographic microscopy can be particularly suitable for mitochondrial trafficking studies, allowing a label-free alternative, ensuring to capture the behavior of mitochondria at high frequencies and for unlimited periods of time, without experimental artefacts. Of course, it is not a direct visualization of the cytoskeleton but it is a simple and indirect measure of its functioning via one of its essential tasks: mitochondrial motility. On the other hand, the absence of specific probes could make harder to clearly identify subcellular structures (unless they are easily recognizable, such as lipid droplets). In Figure 1 we presented selected time frames of supplementary video 1 that we obtained as representative images of a physiological mitochondrial motility obtained with this approach. Mitochondria (shown in bright white) were recognised via their RI and followed (e.g., tracked) over time and space in our sample."
Reviewer 2 Report
The manuscript, "Neurons: the vital and lively interplay between cytoskeleton and ion channels/transporters", aims to review the current literature regarding the interplay between the cytoskeleton, and specifically microtubules, with various cellular functions of neurons, including mobilization of molecular cargo, neuronal migration, neurite outgrowth and synaptic activity. The current manuscript, however, does not cover the topics outlined in the abstract, and the topics that are covered are superficial without adding to the scientific conversation.
1) The first section outlines neurons in a very basic way, but doesn't add anything to the discourse.
2) The second section gives a general explanation of the different types of cytoskeletal elements in neurons and ends with the potential role of cytoskeleton in mitochondrial location, however, there was little detail about the studies that were done to investigate this or what they found. Where do mitochondria need to be within the neurons? What happens if mitochondria are not there? Which elements have been investigated in association with mitochondrial localization within each cellular compartment? Are there other molecular cargo that are involved with cytoskeleton and neuronal function?
3) In the third section the authors touch on the interplay between the cytoskeleton, specifically actin, and the organization of ion channels. There, however, remains a lot that was not included in this realm. How are the different cytoskeleton proteins associated with the Na and K channels at the node and axon initial segment? What studies have been done to investigate the links between cytoskeleton and ion channel clustering and what happens when this doesn't occur properly or is pathologically damaged. How do microtubules play a role in ion channel clustering?
4) In the fourth section the authors begin to discuss the uses of different microscopic techniques to observe cytoskeletal elements, however, they don't discuss how each of these techniques has been used in the investigation of cytoskeletal morphology and what people were able to find about the different cytoskeletal element structures with the link on how this informed their key role in neuronal function. The authors also don't discuss immunohistochemistry/immunocytochemistry, the various antibodies used to assess cytoskeleton, or transgenic technologies.
5) The authors finish Section 4 regarding morphological approaches to visualize and assess cytoskeleton with a discussion about mitochondrial tracking, which is out of place. Additionally, the only figure in this review depicts mitochondrial tracking and not cytoskeleton.
6) In the fifth section the authors use the possible role of mutated EFHC1 leading to mitotic spindle issues in polymicrogyria as strong evidence for the role of cytoskeleton in neuronal function, however, there was no discussion on the potential link to Ca2+ signaling in the cell or how this relates to neuronal cytoskeleton.
7) BDNF is not a cytoskeletal protein and is written as though it is. BDNF does influence cytoskeleton, however, that was not discussed.
8) The authors begin to discuss findings of Xu et al, however, don't explain what the microtubule modulating agent was or how it modulated microtubules.
9) The section on neuropathy did have the level of detail that was expected for the entire review.
Reviewer 3 Report
In the review “Neurons: the vital and lively interplay between cytoskeleton and ion channels/transporters” written by Alberti et al., that was submitted for publication to cells the authors aim to describe our current understanding how components of the cytoskeleton influence neuronal function. Given the fast advance in our understanding on the impact of cytoskeletal dynamics on neuronal function this topic is timely and should be interesting for a broader readership.
The description of the recent advances, however, remains superficial and the direct causal link to the neuronal dysfunction is in most cases not or not properly described. Here, for example the description of how interaction of components of the cytoskeleton with ion channels and transporters – which should be the main topic to the review- is only superficially described in the lined 115 to 123. Here a much more detailed description of the experimental findings including a summarized discussion is currently lacking and needs to be incorporated during revision.
In the following para the authors describe various imaging techniques that for sure have fostered the advancement of our understanding of cytoskeletal structures and more recently also in cytoskeletal dynamics. Although this part might be nice for very junior scientists, it appears to be a bit off topic in this review. I would recommend to shorten this part significantly. Alternatively, individual techniques can be described using a precise example, to delineate how this technique has contributed to our understanding of cytoskeletal functions, including a rigorous and detailed description of the limitations of the respective study.
The para human pathology and cytoskeletal dysfunction is in a way nice, however the authors selected a number of diseases (e.g. fxs, DS, ID and others) that are very complex. Here, the link between the molecular alterations that have been describe in patients derived material and or animal models, and the respective disease phenotype is in many cases not clear. Here the cytoskeletal changes that are described here might (or might not) only a consequence but not the reason for the respective disease. Here the authors need to be more cautious with their interpretations or need to implement much more details on the respective findings to substantiate their interpretation. In its current form most readers will miss the point that from models for each of this diseases valuable information on the contribution of the cytoskeleton to neuronal homeostasis can be obtained.
In the last years many studies have been performed that have addressed the link between the cytoskeleton an both protein trafficking an anchoring. Here a graphical summary of the principal findings (e.g. which class of transport proteins are involved in which kind of transport, and how is substrate specificity is obtained) might be helpful.
Round 2
Reviewer 2 Report
In the revised version of the manuscript the authors have addressed many of my previous comments. I do, however, have a few comments.
1) Good section on the different cytoskeleton elements in the neuron and their function in terms of protein trafficking and anchoring.
2) In the mitochondria transport paragraph, it would be helpful to note if this relationship between an organelle and the cytoskeleton is unique to mitochondria. Based on the work in Giant Squid axons, it doesn't appear to be.
3) Good section on the interaction between cytoskeleton and VOC K and Na channels and Neurofascin.
4) It might be helpful for readers to find the information if the title of section 4 was "microscopic approaches to visualize and assess cytoskeleton" since the use of morphological analysis was not discussed in this section.
5)As there is an in-depth discussion about the microscopy techniques used for cytoskeleton assessments, this should be included in the abstract to help readers identify this information as being part of this review.
6) As the trafficking of mitochondria is a large part of this review, this should be included into the abstract to allow readers to more easily find this information. This takes up a more larger portion of this review than does the discussion of ion channel interactions with cytoskeleton.
7) The section on cytoskeleton as potential biomarkers is underdeveloped. This section, should either be expanded to the level of detail afforded the sections on microscopy or mitochonodrial movement, or removed from the review.
Author Response
We are thankful for the thoughtful suggestions. We modified as follows.
In the revised version of the manuscript the authors have addressed many of my previous comments. I do, however, have a few comments.
1) Good section on the different cytoskeleton elements in the neuron and their function in terms of protein trafficking and anchoring.
Thank you.
2) In the mitochondria transport paragraph, it would be helpful to note if this relationship between an organelle and the cytoskeleton is unique to mitochondria. Based on the work in Giant Squid axons, it doesn't appear to be.
Thank you. We added the suggested note at the end of section 2, addressing this point.
3) Good section on the interaction between cytoskeleton and VOC K and Na channels and Neurofascin.
Thank you.
4) It might be helpful for readers to find the information if the title of section 4 was "microscopic approaches to visualize and assess cytoskeleton" since the use of morphological analysis was not discussed in this section.
Thank you. We modified the title as suggested.
5)As there is an in-depth discussion about the microscopy techniques used for cytoskeleton assessments, this should be included in the abstract to help readers identify this information as being part of this review.
Thank you. We include a reference to this in the abstract.
6) As the trafficking of mitochondria is a large part of this review, this should be included into the abstract to allow readers to more easily find this information. This takes up a more larger portion of this review than does the discussion of ion channel interactions with cytoskeleton.
Thank you. We include a reference to this in the abstract. We also modified the title of the paper for the same reason.
7) The section on cytoskeleton as potential biomarkers is underdeveloped. This section, should either be expanded to the level of detail afforded the sections on microscopy or mitochonodrial movement, or removed from the review.
Thank you. We updated this section giving more details of Neurofilaments light chains as a virtuous example of biomarker development on the basis of a sound biological rationale, to guide future similar research in other fields.
Reviewer 3 Report
In the revised version of the manuscript the authors have tried to address the major critics raised in the first version of revision. In part due to contradicting advices from the reviewer, however, they have tried to develop the review in two different directions simultaneously. Therefore the review still lacks a proper focus. Although I think the new para from lines 82-114 is the first in the whole manuscript that provides scientific information exceeding text book knowledge, it still lacks an interpretational view that opens up or at least emphasise new direction of research on cytoskeletal functions in neurons. Here the authors need to focus on questions that arises from the studies they describe in their review. The same critics applies to the new para on ion-channel cytoskeletal interactions (lines 138-177)
The description of the different light microscopic techniques, does, at least from my point of view, not fit into this review. This is to a large extend text book knowledge that distracts the attention from the major topic of this review, that should be given the title – the function of the cytoskeleton in neuron. Therefore this para should be removed of significantly shortened.
In chapter 5 the authors describe potential links between human pathology and malfunctions of the cytoskeleton. This para should be the central core of the manuscript and must be extended. Here, an interpretational view on previous research, that allows the reader to understand the bigger picture of research on cytoskeletal functions in neurons is missing completely and need to be incorporated.
Moreover, there are still a lot of typos and grammatical errors in the manuscript that require a rigorous language editing. Although I see the potential of the topic, in its current version the review does not contribute significantly to the scientific discourse and therefore should be extensively revised (which means that large parts have to be rewritten and restructured) before considering it again for publication.
Author Response
We are thankful for the thoughtful suggestions. We modified as follows.
In the revised version of the manuscript the authors have tried to address the major critics raised in the first version of revision. In part due to contradicting advices from the reviewer, however, they have tried to develop the review in two different directions simultaneously. Therefore the review still lacks a proper focus. Although I think the new para from lines 82-114 is the first in the whole manuscript that provides scientific information exceeding text book knowledge, it still lacks an interpretational view that opens up or at least emphasise new direction of research on cytoskeletal functions in neurons. Here the authors need to focus on questions that arises from the studies they describe in their review. The same critics applies to the new para on ion-channel cytoskeletal interactions (lines 138-177)
We are thankful for this comments. It was our intention to give an overview that can be relevant also for less experienced readers. In this part, we lay the ground for the general understanding of the basic mechanism: we better specified this and we demonstrated how to use this information for novel and future research studies in section 5 which was also revised as asked.
The description of the different light microscopic techniques, does, at least from my point of view, not fit into this review. This is to a large extend text book knowledge that distracts the attention from the major topic of this review, that should be given the title – the function of the cytoskeleton in neuron. Therefore this para should be removed of significantly shortened.
We are thankful for this comments. However, this section was prepared according to our Reviewers’ suggestion. Once again, we aim at a broader readership and, therefore, we fill this paragraph might be useful to young investigator. As we better explained in the text, it gives an overview of some techniques (pros and cons), older and newer, to be used in research study designing.
In chapter 5 the authors describe potential links between human pathology and malfunctions of the cytoskeleton. This para should be the central core of the manuscript and must be extended. Here, an interpretational view on previous research, that allows the reader to understand the bigger picture of research on cytoskeletal functions in neurons is missing completely and need to be incorporated.
We are thankful for this comments. We better clarified this paragraph and its scope. We present 3 main human diseases (epilepsy, intellectual disability and peripheral neuropathies) and we explain how in the past cytoskeleton was linked to this condition (in most cases its role is still not completely and extensively investigated) to provide suggestion to drive future translational research projects.
Moreover, there are still a lot of typos and grammatical errors in the manuscript that require a rigorous language editing. Although I see the potential of the topic, in its current version the review does not contribute significantly to the scientific discourse and therefore should be extensively revised (which means that large parts have to be rewritten and restructured) before considering it again for publication.
Thank you. An English-native speaker revised the final draft.